# The transcriptional regulator BZR1 mediates trade-off between plant innate immunity and growth

**Rosa Lozano-Durán[1], Alberto P Macho[1], Freddy Boutrot[1], Cécile Segonzac[1†], Imre E Somssich[2], Cyril Zipfel[1]\***

[1]The Sainsbury Laboratory, Norwich, United Kingdom; [2]Max Planck Institute for Plant Breeding Research, Köln, Germany

**Abstract** The molecular mechanisms underlying the trade-off between plant innate immunity and steroid-mediated growth are controversial. Here, we report that activation of the transcription factor BZR1 is required and sufficient for suppression of immune signaling by brassinosteroids (BR). BZR1 induces the expression of several WRKY transcription factors that negatively control early immune responses. In addition, BZR1 associates with WRKY40 to mediate the antagonism between BR and immune signaling. We reveal that BZR1-mediated inhibition of immunity is particularly relevant when plant fast growth is required, such as during etiolation. Thus, BZR1 acts as an important regulator mediating the trade-off between growth and immunity upon integration of environmental cues.

**\*For correspondence:** cyril.
zipfel@tsl.ac.uk

**†Present address:** Institute of Agriculture and Environment, Massey University Manawatu, Palmerston North, New Zealand

**Competing interests:** The authors declare that no competing interests exist.

## Introduction

The trade-off between plant growth and immunity needs to be finely regulated to ensure proper allocation of resources in an efficient and timely manner upon effective integration of environmental cues (*Pieterse et al., 2012*). A key aspect of plant immunity is the perception of pathogen-associated molecular patterns (PAMPs) by surface-localized pattern-recognition receptors (PRRs), leading to PAMP-triggered immunity (PTI) (*Dodds and Rathjen, 2010*). PRRs of the leucine-rich repeat receptor kinases (LRR-RKs) class rely on the regulatory LRR-RK BAK1 (BRASSINOSTEROID INSENSITIVE 1-ASSOCIATED KINASE 1) for signaling (*Monaghan and Zipfel, 2012*); that is the case of FLS2 (FLAGELLIN SENSITIVE 2) and EFR (EF-TU RECEPTOR), which perceive bacterial flagellin (or the active peptide flg22) and EF-Tu (or the active peptide elf18) respectively. BAK1 also interacts with the LRR-RK BRI1 (BRASSINOSTEROID INSENSITIVE 1), the main receptor for the growth-promoting steroid hormones brassinosteroids (BR), and is a positive regulator of BR-mediated growth (*Kim and Wang, 2010*). Hence, a crosstalk between the BR- and PAMP-triggered signaling pathways resulting from competition for BAK1 was hypothesized. While a unidirectional antagonism between BR and PTI signaling has been recently described in Arabidopsis (*Albrecht et al., 2012*; *Belkhadir et al., 2012*), the exact underlying mechanisms are still controversial. Activation of the BR signaling pathway via either transgenic overexpression of *BRI1* or the BR biosynthetic gene *DWF4* or expression of the activated BRI1 allele *BRI1sud* suppresses PTI outputs (*Belkhadir et al., 2012*). One such output, the PAMP-triggered callose deposition, could be restored by over-expression of *BAK1-HA*, suggesting that BAK1 is a limiting factor (*Belkhadir et al., 2012*). However, exogenous BR treatment of wild-type plants does not affect the FLS2-BAK1 complex formation upon FLS2 activation, while it results in decreased PTI responses (*Albrecht et al., 2012*).

## Results and discussion

In order to clarify the role of BAK1 in the BR-PTI crosstalk, we investigated FLS2-BAK1 complex formation in the transgenic Arabidopsis lines overexpressing *BRI1* or *DWF4* or expressing *BRI1sud* (*Belkhadir et al.,*

**eLife digest** Like all organisms, plants must perform a careful balancing act with their resources. Investing in the growth of new roots or leaves can allow a plant to better exploit its environment—but it must not be at the expense of leaving the plant vulnerable to attack by pests and pathogens. As such, there is an obvious trade-off between allocating resources to growth or defense against disease. This trade-off must be finely balanced, and must also be responsive to different cues in the environment that would favor either growth or defense.

The plant's immune system is able to detect invading microbes, and trigger a defensive response against them. At the surface of plant cells, proteins called pattern recognition receptors are able to recognize specific molecules that are the tell-tale signs of microbes and pathogens—such as the proteins in the molecular tails that bacteria use to move around.

For many pattern recognition receptors, signaling that they have recognized a potential invading microbe requires the actions of a co-receptor called BAK1. Interestingly, BAK1 also interacts with the receptor that identifies brassinosteroids—hormones that stimulate plant growth. Since growth and a functioning immune system are both reliant on BAK1, it was hypothesized that competition for this co-receptor could have a role in the trade-off between the two processes in plants. However, this explanation was controversial and the mechanisms underlying the trade-off still required clarification.

Now, Lozano-Durán et al. have debunked the idea that competition for BAK1 is directly responsible for the trade-off between growth and immunity. By examining how BAK1 interacts with immune receptors in the plant model species *Arabidopsis thaliana*, the trade-off was actually shown to be independent of BAK1. Instead, it was discovered that activation of a protein, called BZR1, reprogramed gene expression to 'switch off' immune signaling in response to brassinosteroids.

Lozano-Durán et al. also show that BZR1 allows the balance of the trade-off between growth and immunity to be shifted in response to cues from the environment. The suppression of the immune system by BZR1 was particularly pronounced when the conditions required fast plant growth—for example, when they mimicked the conditions experienced by seedlings before they emerge from the soil, and must grow swiftly to reach the light before they starve.

*2012*). Upon treatment with flg22, FLS2 associated normally with BAK1 in these transgenic plants, and neither FLS2 nor BAK1 accumulation was altered (*Figure 1—figure supplement 1A*). Moreover, these plants displayed a weaker reactive oxygen species (ROS) burst in response to chitin (*Figure 1—figure supplement 1B*), whose signaling pathway is BAK1-independent (*Shan et al., 2008*; *Ranf et al., 2011*). This result is consistent with the previous finding that exogenous BR treatment can also inhibit the chitin-induced ROS burst (*Albrecht et al., 2012*). BAK1-HA is not fully functional in BR signaling and exerts a dominant-negative effect on the endogenous BAK1 (*Figure 1—figure supplement 1C*), which may explain that introduction of the *BAK1-HA* transgene can override the suppression of immunity triggered by overexpression of *BRI1* (*Belkhadir et al., 2012*); BAK1-HA does not exert such a dominant negative effect, however, on PTI signaling (*Figure 1—figure supplement 1D*). Taken together, these results indicate that the BR-mediated suppression of PTI is triggered independently of a competition between BRI1 and PRRs for BAK1.

We sought to determine at which level of the BR signaling pathway the antagonism initiates. After BR perception by BRI1 and activation of the BRI1-BAK1 complex, the BR signal transduction cascade includes inactivation of BIN2 (BR INSENSITIVE 2) and BIN2-like kinases, a family of GSK3-like kinases acting as negative regulators of the pathway (*Vert and Chory, 2006*). This leads to dephosphorylation of BZR1 (BRASSINAZOLE RESISTANT 1) and BES1/BZR2 (BRI1-EMS-SUPPRESSOR 1/BRASSINAZOLE RESISTANT 2), two bHLH transcription factors acting as major regulators of BR-induced transcriptional changes, which then become active (*Wang et al., 2002*; *Yin et al., 2002*). Treatment with the chemicals LiCl and bikinin, which inhibit GSK3-like kinases (*De Rybel et al., 2009*; *Yan et al., 2009*), resulted in impaired flg22-triggered ROS burst (*Figure 1A,B*), as observed upon genetic or ligand-induced activation of the BR pathway. Furthermore, a triple mutant in *BIN2* and the two closest related GSK3-like kinases, *BIL1* (*BIN2-LIKE 1*) and *BIL2* (triple GSK3 mutant; *Vert and Chory, 2006*), shows a similar impairment in response to either flg22 or chitin (*Figure 1C*). Interestingly, in spite of regulating MAPKs involved in

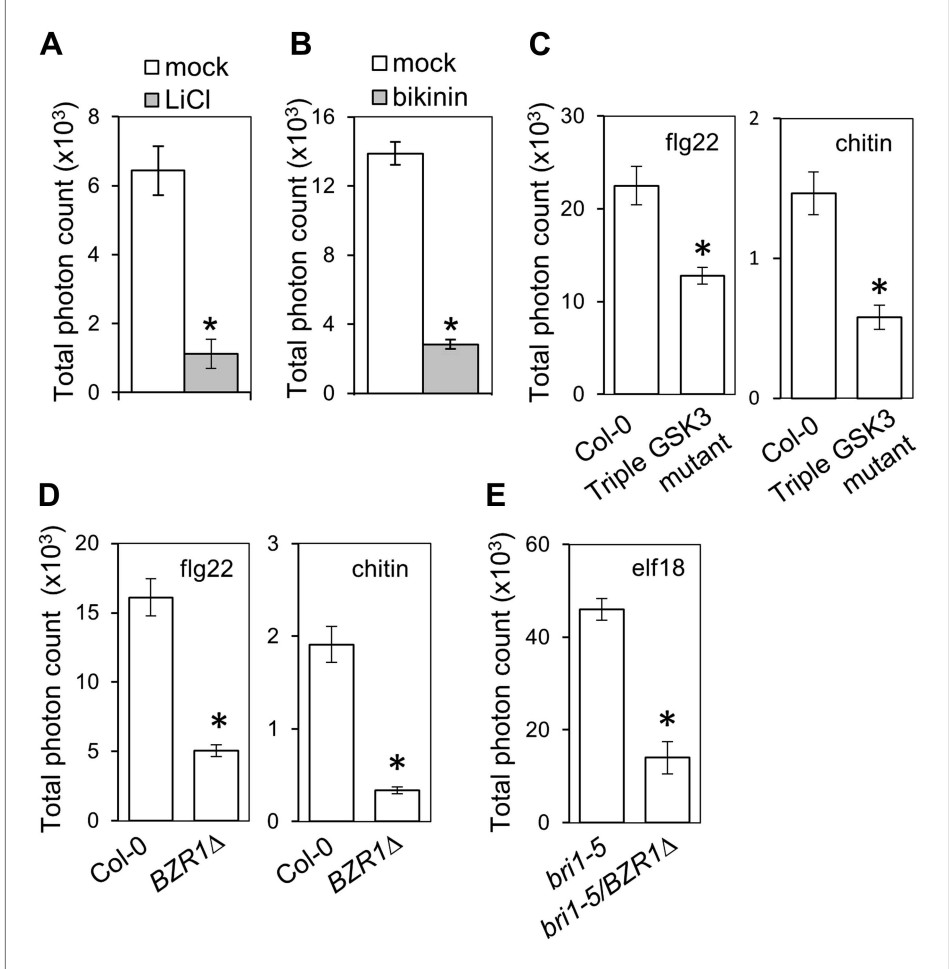

**Figure 1**. Activation of BZR1 is sufficient to inhibit the PAMP-triggered ROS burst. (**A**) and (**B**) Flg22-triggered ROS burst after LiCl (**A**) or bikinin (**B**) treatment. Leaf discs were pre-treated with a 10 mM LiCl solution for 5 hr or with a 50 µM bikinin solution for 16 hr. (**C**) Flg22- or chitin-triggered ROS burst in Col-0 and the triple GSK3 mutant plants. (**D**) Flg22- or chitin-induced ROS burst in Col-0 and *BZR1Δ* plants. (**E**) Elf18-triggered ROS burst in *bri1-5* and *bri1-5/BZR1Δ* plants. In all cases, bars represent SE of n = 28 rosette leaf discs. Asterisks indicate a statistically significant difference compared to the corresponding control (mock treatment [**A** and **B**], Col-0 [**C** and **D**] or *bri1-5* [**E**]), according to a Student's *t*-test (p<0.05). Leaf discs of four- to five-week-old Arabidopsis plants were used in these assays. Flg22 and elf18 were used at a concentration of 50 nM; chitin was used at a concentration of 1 mg/ml. Total photon counts were integrated between minutes two and 40 after PAMP treatment. All experiments were repeated at least three times with similar results.

The following figure supplements are available for figure 1:

**Figure supplement 1**. The BR-mediated suppression of PTI can be triggered independently of a competition for BAK1.

**Figure supplement 2**. PAMP-triggered MAPK activation is not impaired upon activation of BR signaling.

**Figure supplement 3**. Activation of BZR1, but not BES1, is sufficient to inhibit the PAMP-triggered ROS burst.

stomata development (*Kim et al., 2012*; *Khan et al., 2013*), neither BR treatment nor loss of function of BIN2 affect flg22-triggered MAPK activation (*Figure 1—figure supplement 2*), contrary to what has been recently suggested (*Choudhary et al., 2012*; *Zhu et al., 2013*). These results indicate that the BR-PTI crosstalk occurs downstream of BIN2.

Transgenic expression of two different constitutively active versions of BZR1, BZR1Δ (*Gampala et al., 2007*) and BZR1S173A (*Ryu et al., 2007*), results in impaired flg22- or chitin-triggered ROS burst

(*Figure 1D*, *Figure 1—figure supplement 3A*). Consistent with previous results (*Albrecht et al., 2012*; *Figure 1—figure supplements 1A and 2*), plants expressing *BZR1Δ* or *BZR1^S173A* display normal FLS2-BAK1 complex formation and MAPK activation upon flg22 treatment (*Figure 2A,B*, *Figure 2—figure supplement 1A,B*), but are impaired in PAMP-triggered marker gene expression, seedling growth inhibition (SGI) (*Figure 2C–E*) and induced resistance to *P. syringae* pv. *tomato* (*Pto*) DC3000

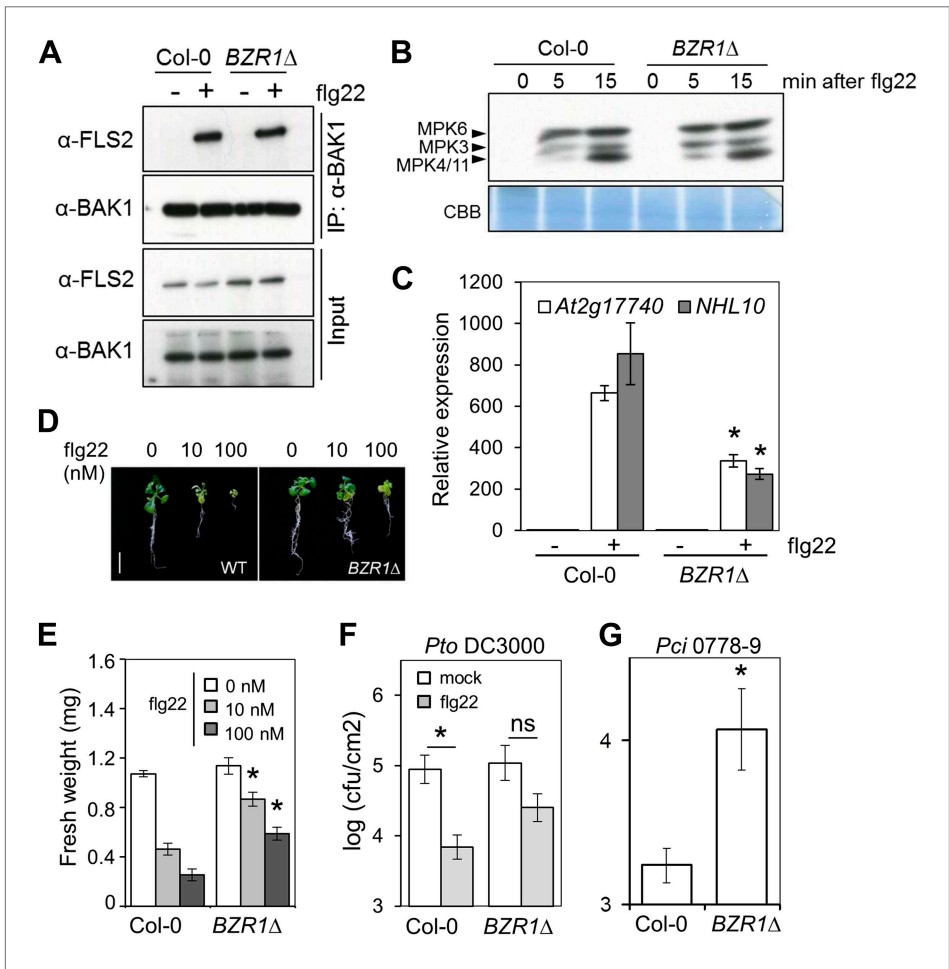

**Figure 2**. Activation of BZR1 results in the suppression of specific PTI outputs. (**A**) Co-immunoprecipitation (Co-IP) of BAK1 and FLS2 in Col-0 and *BZR1Δ* seedlings after 10 min mock (−) or 1 μM flg22 (+) treatment. Proteins were separated in a 10% acrylamide gel and transferred to PVDF membranes. Membranes were blotted with anti-FLS2 or anti-BAK1 antibodies. (**B**) MAPK activation in Col-0 and *BZR1Δ* seedlings upon 1 μM flg22 treatment. Proteins were separated in a 10% acrylamide gel and transferred to PVDF membranes. Membranes were blotted with phospho-p44/42 MAPK (Erk1/2; Thr202/Tyr204) rabbit monoclonal antibodies. CBB: Coomassie brilliant blue. (**C**) Marker gene (*At2g17700* and *NHL10*) expression in Col-0 and *BZR1Δ* seedlings after 1 hr mock (−) or 1 μM flg22 (+) treatment, as determined by qPCR. Bars represent SE of n = 3. (**D**) and (**E**) Seedling growth inhibition of 10-day-old Col-0 or *BZR1Δ* seedlings induced by increasing concentrations of flg22, as indicated. Scale bar (**D**), 1 cm. Bars (**E**) represent SE of 8 ≤ n ≤ 16. (**F**) Flg22-induced resistance to *P. syringae* pv. *tomato* DC3000 in Col-0 and *BZR1Δ* plants. Plants were pre-treated with 1 μM flg22 or water 24 hr prior to bacterial infiltration. Bars represent SE of n = 4. This experiment was repeated seven times with similar results. (**G**) Susceptibility of Col-0 and *BZR1Δ* plants to *P. syringae* pv. *cilantro* 0788-9. Bars represent SE of n = 4. Asterisks indicate a statistically significant difference compared to Col-0 according to a Student's *t*-test (p<0.05); ns = not significant. All experiments were repeated at least twice with similar results unless otherwise stated.

The following figure supplements are available for figure 2:

**Figure supplement 1**. Expression of the constitutively active BZR1^S173A results in the suppression of specific PTI outputs.

(*Figure 2F*, *Figure 2—figure supplement 1C*), and are more susceptible to the non-host strain *Pseudomonas syringae* pv. *cilantro* (*Pci*) 0788-9 (*Lewis et al., 2010*) (*Figure 2G*). Notably, transgenic expression of a constitutively active form of BES1, BES1[S171A] (*Gampala et al., 2007*), does not impact the flg22-triggered ROS burst (*Figure 1—figure supplement 3B*). We then tested if activation of BZR1 is sufficient to inhibit PTI signaling. Induction of BR signaling by bikinin treatment still represses elf18-induced ROS burst in the BRI1 mutant *bri1-5* (we used elf18 in this experiment because *bri1-5* is in the Ws-2 background, which is a natural *fls2* mutant) (*Figure 1—figure supplement 3C*). *bri1-5/BZR1Δ* plants (*Gampala et al., 2007*) still exhibited reduced PAMP-triggered ROS burst (*Figure 1E*), and treatment with the BR biosynthetic inhibitor brassinazole (BRZ) did not affect the *BZR1Δ* effect (*Figure 1—figure supplement 3D*). Interestingly, BRZ treatment of wild-type Col-0 plants results in increased ROS production (*Figure 1—figure supplement 3D*), which is consistent with the fact that BR inhibits PTI responses and suggests that endogenous concentrations of the hormone exert this effect. These results demonstrate that activation of BZR1 affects PTI signaling independently of BR perception or synthesis.

To understand how BZR1 mediates the BR-PTI crosstalk, we performed meta-analysis of microarray and ChIP-chip data containing BR-regulated and BZR1 or BES1 target genes (*Sun et al., 2010*; *Yu et al., 2011*). Functional enrichment of BR-regulated genes unveiled a statistically significant over-representation of defense-related GO terms of the Biological Process ontology (*Table 1*), indicating that BR signaling regulates the expression of defense-related genes. Independent analysis of BR-regulated BZR1 or BES1 targets confirmed BZR1 as the main transcription factor involved in the regulation of defense gene expression (*Table 1*). Two out of four over-represented GO terms of the Molecular Function ontology among the BR-regulated BZR1 targets are transcription factor and transcription repressor activity (*Table 2*). Interestingly, several defense-related GO terms are also over-represented in the subset of BR-regulated BZR1-targeted transcription factors (*Table 3*), pointing at a BZR1-mediated secondary transcriptional wave of defense-related genes.

To identify BZR1-regulated transcription factors with a prominent role in defense, we performed promoter enrichment analysis on the subset of defense-related BR-regulated genes, and found the W-box motif as the only significantly over-represented motif (*Table 4*). The W-box motif is the binding site for the WRKY family of transcription factors (*Rushton et al., 2010*), and several members of this family are BR-regulated BZR1-targets (*Table 5*). We hypothesized that WRKYs that are BR-induced and BZR1 targets may be involved in PTI signaling. Notably, *wrky11*, *wrky15*, *wrky18* and *wrky70* mutants displayed enhanced PAMP-triggered ROS (*Figure 3A*), suggesting that these transcription factors

**Table 1.** Defense-related Gene Ontology terms (Biological Process ontology) over-represented among all BR-regulated genes, BR-regulated BZR1 targets and BR-regulated BES1 targets

| Defense-related GO term | Observed frequency (%) | Expected frequency (%) | p-value |
| --- | --- | --- | --- |
| BR-Regulated genes | | | |
| response to bacterium | 2.2 | 1 | $3.31 \times 10^{-08}$ |
| defense response to bacterium | 1.9 | 0.8 | $3.31 \times 10^{-08}$ |
| response to chitin | 1.4 | 0.5 | $1.78 \times 10^{-07}$ |
| defense response | 4.7 | 3 | $3.32 \times 10^{-07}$ |
| response to fungus | 1.5 | 0.7 | $3.4 \times 10^{-06}$ |
| response to nematode | 0.7 | 0.2 | 0.000532 |
| defense response to fungus | 1 | 0.5 | 0.0035 |
| BR-regulated BZR1 targets | | | |
| response to chitin | 2.6 | 0.5 | $9.13 \times 10^{-13}$ |
| response to bacterium | 2.3 | 1 | 0.00112 |
| defense response to bacterium | 1.9 | 0.8 | 0.00154 |
| response to fungus | 1.6 | 0.7 | 0.00495 |
| BR-regulated BES1 targets | | | |
| response to chitin | 2.4 | 0.5 | 0.00439 |

**Table 2.** Gene Ontology terms (Molecular Function ontology) over-represented among all BR-regulated BZR1 targets

| Over-represented GO term | Observed frequency (%) | Expected frequency (%) | p value |
|---|---|---|---|
| BR-regulated BZR1 targets | | | |
| nucleic acid binding transcription factor activity | 14.8 | 10.2 | 0.000223 |
| transferase activity | 21.6 | 16.8 | 0.00333 |
| kinase activity | 11.6 | 8.1 | 0.00702 |
| transcription repressor activity | 1.1 | 0.3 | 0.01 |

act as negative regulators of early PTI signaling. This is in accordance with their role as negative regulators of immunity (*Figure 3—figure supplement 1A*; *Journot-Catalino et al., 2006*). Therefore, the BZR1-mediated inhibition of PTI might be partially explained by the up-regulation of genes encoding WRKY transcription factors that negatively control the expression of genes involved in early PTI signaling.

One of the *WRKY* genes targeted by BZR1 is *WRKY40* (*Sun et al., 2010*). Interestingly, all described targets of WRKY40 (*Pandey et al., 2010*) are also targets of BZR1 (*Sun et al., 2010*) (*Table 6*). The over-representation of the W-box motif among BZR1 targets (*Table 7*) suggests that BZR1 may interact with WRKY transcription factors (such as WRKY40) to cooperatively regulate transcription. *WRKY40* has been described as a negative regulator of defense against biotrophic pathogens and insects (*Xu et al., 2006*; *Pandey et al., 2010*; *Brotman et al., 2013*; *Schon et al., 2013*; *Schweizer et al., 2013*). In agreement with this, we found that a null *wrky40* mutant is more resistant to *Pto* DC3000 (*Figure 3—figure supplement 1B*). Strikingly, *wrky40* plants are partially impaired in the BR-mediated suppression of PAMP-triggered ROS (*Figure 3B*), suggesting that WRKY40 may act coordinately with BZR1 to suppress immunity. Indeed, we found that BZR1 associates with WRKY40, but not WRKY6, in co-immunoprecipitation experiments when transiently co-expressed in *Nicotiana benthamiana* leaves (*Figure 3C*) or Arabidopsis protoplasts (*Figure 3D*). Collectively, these results indicate that BZR1 and WRKY40 form a protein complex that may participate in the transcriptional inhibition of PTI signaling.

BZR1, together with DELLAs and PIF4, is part of a core transcription module that integrates hormonal (gibberellin [GA] and BR) and environmental (light) signals (*Gallego-Bartolome et al., 2012*; *Li et al., 2012*; *Oh et al., 2012*; *Bai et al., 2012b*). In the dark, BZR1 is activated by endogenous BR and GA to promote growth, partially through the synergistic interaction with PIF4 (*Jaillais and Vert, 2012*).

**Table 3.** Defense-related Gene Ontology terms (Biological Process ontology) over-represented among the BZR1-target BR-regulated transcription factors

| Defense-related GO Term | Observed frequency (%) | Expected frequency (%) | p value |
|---|---|---|---|
| BZR1-target BR-regulated TFs | | | |
| response to chitin | 16.6 | 0.5 | $1.36 \times 10^{-26}$ |
| defense response to bacterium | 7.6 | 0.8 | $4.71 \times 10^{-07}$ |
| response to bacterium | 7.6 | 1 | $4.51 \times 10^{-06}$ |
| regulation of defense response to virus by host | 1.4 | 0 | 0.000964 |
| regulation of immune effector process | 1.4 | 0 | 0.00151 |
| regulation of defense response to virus | 1.4 | 0 | 0.00151 |
| regulation of defense response | 2.8 | 0.3 | 0.00484 |
| defense response | 8.3 | 3 | 0.00603 |
| response to fungus | 3.4 | 0.7 | 0.01 |
| defense response to fungus | 2.8 | 0.5 | 0.02 |

**Table 4.** Over-represented *cis*-acting promoter elements among the defense-related BR-regulated genes according to Athena (http://www.bioinformatics2.wsu.edu/cgi-bin/Athena/cgi/home.pl)

| Enriched TF site | % promoters | p value |
|---|---|---|
| Defense-related BR-regulated genes | | |
| W-box | 72.4 | $<10^{-6}$ |

Because etiolation requires rapid growth, we hypothesized that plants may prioritize growth over immunity in dark conditions. In fact, we found that PAMP-triggered SGI was partially impaired in dark-grown seedlings (*Figure 4A–D*). This impairment was abolished in the BR-insensitive mutants *bri1-301* and *bin2-1* (*Figure 4A*, *Figure 4—figure supplement 2A*), indicating that BR signaling is responsible for the dark-induced suppression of this PTI response. Activation of BZR1 in the *BZR1Δ* line mimicked the dark-induced suppression of SGI in both light and dark (*Figure 4B*). However, activation of BES1 in the *BES1^{S171A}* line did not impact SGI (*Figure 4—figure supplement 3A*). Consistent with the previous results, exogenous BR treatment suppressed SGI in both light and dark (*Figure 4C*, *Figure 4—figure supplement 2B,C*). While treatment with GA alone did not have a dramatic effect on SGI, co-treatment with BL and GA resulted in an enhancement of the BR-mediated suppression of SGI (*Figure 4C*, *Figure 4—figure supplement 2B*), indicating an additive effect of these two hormones when applied together. Moreover, treatment with the GA synthesis inhibitors paclobutrazol (PAC) or uniconazole (Uni) abolished the effect of BL on SGI (*Figure 4—figure supplement 1A,B* and *Figure 4—figure supplement 2B,C*), and this effect was reduced in the GA biosynthetic mutant *ga1-3* (*Figure 4—figure supplement 1C*). Taken together, these results demonstrate that BR suppress at least one PTI output, SGI, in the dark in a GA-dependent manner, most likely through activation of BZR1. Notably, although the *wrky40* mutant undergoes etiolation normally (*Figure 4—figure supplement 2D*), it shows a diminished suppression of SGI in the dark (*Figure 4D*, *Figure 4—figure supplement 2D*), supporting the idea that WRKY40 is required for the BZR1-mediated inhibition of PTI.

Previously, a unidirectional negative crosstalk between the growth-promoting hormone BR and PTI had been described (*Albrecht et al., 2012*; *Belkhadir et al., 2012*). In this work, we show that activation of one of two major BR-activated transcription factors, BZR1, is sufficient to suppress PTI, measured as PAMP-triggered ROS production, PAMP-triggered gene expression, SGI or induced resistance (*Figures 1 and 2*, *Figure 1—figure supplement 3*, *Figure 2—figure supplement 1*). Of note, another PTI output, MAPK activation, is not affected by activation of the BR pathway (*Figure 2B*, *Figure 1—figure supplement 2*, *Figure 2—figure supplement 1B*). BR treatment results in BZR1-dependent changes in the expression of defense-related genes, among which several members of the WRKY family of transcription factors can be found. Because the promoters of BR-regulated defense-related genes are enriched in the W-box motif (*Table 4*), BZR1-targeted *WRKY* transcription factors could be responsible for a secondary wave of transcription, ultimately leading to the suppression of PTI. In agreement with this idea, a subset of *WRKY*s induced by BR (*WRKY11*, *WRKY15* and *WRKY18*) (*Figure 3A*) act as negative regulators of PAMP-triggered ROS, potentially by controlling the steady-state expression of genes encoding components required for this response. The overrepresentation of the W-box motif among the BZR1 targets (*Table 7*) raises the possibility that, additionally, WRKY transcription factors could also act together with BZR1 to cooperatively regulate gene expression. We found that WRKY40 associates with BZR1 directly or indirectly in planta

**Table 5.** BR-regulated BZR1-target *WRKY* genes

| AGI number | WRKY TF |
|---|---|
| BR-Induced BZR1 targets | |
| AT4G31800 | WRKY18 |
| AT4G31550 | WRKY11 |
| AT4G23810 | WRKY53 |
| AT3G56400 | WRKY70 |
| AT5G49520 | WRKY48 |
| AT5G52830 | WRKY27 |
| AT1G69310 | WRKY57 |
| AT2G23320 | WRKY15 (*Yu et al., 2011*) |
| BR-repressed BZR1 targets | |
| AT4G01250 | WRKY22 |
| AT1G80840 | WRKY40 |
| AT2G24570 | WRKY17 |
| AT2G23320 | WRKY15 (*Sun et al., 2010*) |
| AT2G30590 | WRKY21 |

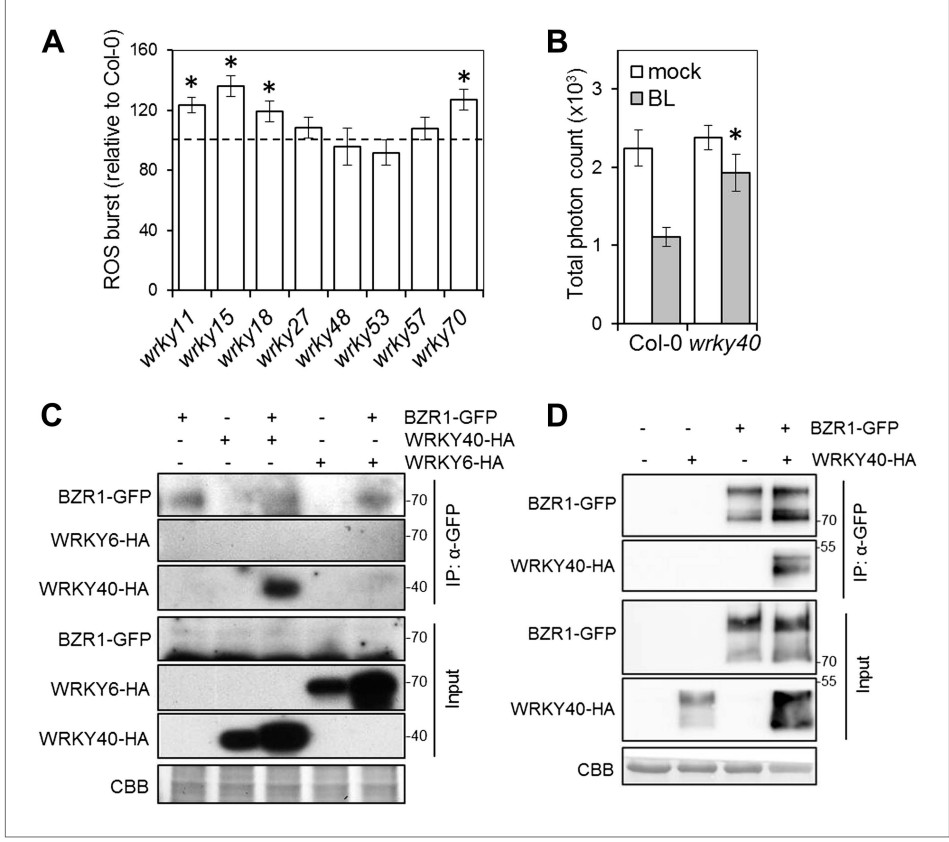

**Figure 3**. WRKY transcription factors play a dual role on the BR-mediated regulation of PTI signaling. (**A**) Flg22-triggered ROS burst in mutants in each BR-induced BZR1-targeted *WRKY*. Leaf discs of four- to five-week-old Arabidopsis plants were used in these assays. Flg22 was used at a concentration of 50 nM. Total photon counts were integrated between minutes two and 40 after PAMP treatment. Bars represent SE of n = 28. Asterisks indicate a statistically significant difference compared to Col-0 according to a Student's *t*-test (p<0.05). (**B**) Flg22-triggered ROS burst in epiBL (BL)- or mock- pre-treated *wrky40* mutant or wild-type plants. Leaf discs of four- to five-week-old plants were pre-treated with a 1 μM BL solution or mock solution for 8 hr. Flg22 was used at a concentration of 50 nM. Total photon counts were integrated between minutes two and 40 after PAMP treatment. Bars represent SE of n = 21. Asterisks indicate a statistically significant difference compared to Col-0 according to a Student's *t*-test (p<0.05). (**C**) Co-IP of BZR1-GFP transiently expressed in *N. benthamiana*, alone or together with WRKY40-HA or WRKY6-HA. BZR1-GFP was immunoprecipitated with an anti-GFP antibody. Immuniprecipitated or total proteins were separated in a 10% acrylamide gel and transferred to PVDF membranes. Membranes were blotted with anti-HA or anti-GFP antibodies. CBB: Coomassie brilliant blue. (**D**) Co-IP of BZR1-GFP transiently expressed in Arabidopsis protoplasts, alone or together with WRKY40-HA. BZR1-GFP was immunoprecipitated with an anti-GFP antibody. Immuniprecipitated or total proteins were separated in a 10% acrylamide gel and transferred to PVDF membranes. Membranes were blotted with anti-HA or anti-GFP antibodies. CBB: Coomassie brilliant blue. All experiments were repeated at least twice with similar results.

The following figure supplements are available for figure 3:

**Figure supplement 1**. Mutants in *WRKY11*, *WRKY15*, *WRKY18* and *WRKY40* are more resistant to *Pto* DC3000.

---

(*Figure 3C,D*); in the absence of WRKY40, the BR-mediated suppression of PAMP-triggered ROS burst is partially impaired (*Figure 3B*). Therefore, WRKYs may play a dual role in the BZR1-mediated suppression of defense, as both co- and secondary regulators of defense gene expression. Given that the loss of BR-mediated suppression of PAMP-triggered ROS burst in the *wrky40* mutant is only partial, BZR1 may interact with other members of the WRKY family, such as WRKY18 or WRKY60, to repress immunity.

Furthermore, we recently described that the bHLH transcription factor HBI1, which is a BRZ1 target (*Sun et al., 2010*; *Bai et al., 2012a*), negatively regulates PTI (Malinovsky et al., under revision). All together, these results illustrate that BZR1 controls the expression of transcription factors (e.g. WRKY11,

**Table 6.** Overlap between the targets of WRKY40 and BZR1

| Known targets of WRKY40 (*Pandey et al., 2010*) | Targets of BZR1 (*Sun et al., 2010*) |
| --- | --- |
| Confirmed by ChIP | |
| EDS1 | Yes |
| RRTF1 | Yes |
| JAZ8 | Yes |
| Putative (according to expression analyses) | |
| LOX2 | Yes |
| AOS | Yes |
| JAZ7 | Yes |
| JAZ10 | Yes |

**Table 7.** Representation of the W-box motif among the BR-regulated BZR1 targets according to Athena (http://www.bioinformatics2.wsu.edu/cgi-bin/Athena/cgi/home.pl)

| BZR1 targets | % of promoters with W-box motif(s) | p value |
| --- | --- | --- |
| BR-induced | 66 | $<10^{-10}$ |
| BR-repressed | 72 | $<10^{-4}$ |

WRKY15, WRKY18 and HBI1), which themselves might control the expression of PTI components (see model in *Figure 4E*) whose identities remain to be identified.

Plants need to finely regulate allocation of resources upon integration of environmental cues, both biotic and abiotic, in order to rapidly and readily adapt to changing conditions and ensure survival in a cost-efficient manner. Dark conditions impose an energetic limitation due to lack of photo-assimilates; in this situation, the restoration of normal photosynthesis by reaching light is an essential requirement to guarantee perpetuation, and as such must be given priority (*Casal, 2013*). We hypothesize that when plants face conditions that require rapid growth, such as when germinating in soil or when under a canopy, limited resources are invested in this developmental process at the expense of immunity in what must be a quantitative choice. Indeed, we show that etiolated seedlings do not arrest their growth in response to PAMPs as light-grown seedlings do, as measured by total fresh weight (*Figure 4A–D*). In addition, BR signaling, acting cooperatively with GA signaling, is required for the dark-induced suppression of this PTI response (*Figure 4C*, *Figure 4—figure supplement 1*), and activation of BZR1 is sufficient to exert this effect regardless of light conditions (*Figure 4B*). Although seedlings were used in these experiments due to technical reasons, BR also regulate growth at later developmental stages, so this phenomenon may be more general. Based on these findings, we propose a model in which BZR1 regulates the expression of defense genes, assisted by WRKY40 (and potentially other WRKYs), ultimately resulting in a quantitative suppression of immunity (*Figure 4E*). Because the activation status of BZR1 depends on BR, GA and light signaling, BZR1 would act as a molecular integrator of these inputs to effectively regulate the trade-off between growth and immunity.

## Materials and methods

### Plant materials and growth conditions

Col-0 plants were used as control. The transgenic lines *BZR1Δ*, *bri1-5/BZR1Δ* and *BES1$^{S171A}$* (*Gampala et al., 2007*), *BZR1$^{S173A}$* and *BZR1-CFP* (*Ryu et al., 2007*), *35S:BRI1-cit*, *BRI1p:BRI1$^{sud}$-cit*, *35S:DWF4* and *BAK1-HA* (*Belkhadir et al., 2012*) are published. The mutant lines Triple GSK3 mutant (*Vert and Chory, 2006*), *bri1-5* (*Noguchi et al., 1999*), *bri1-301* (*Xu et al., 2008*), *bin2-1* (*Peng et al., 2008*), *wrky11* (*Journot-Catalino et al., 2006*), *wrky18*, *wrky53* and *wrky70* (*Wang et al., 2006*) *wrky27* (*Mukhtar et al., 2008*), *wrky40* (*Pandey et al., 2010*) and *ga1-3* (*Navarro et al., 2008*) are published. *wrky15* mutant was identified in the ZIGIA population (*Wisman et al., 1998a, 1998b*); *wrky48* and *wrky57* are from the SALK collection (*Alonso et al., 2003*).

Arabidopsis plants and seedlings were grown as described in *Albrecht et al. (2012)*.

### Chemicals

Flg22 and elf18 peptides were purchased from Peptron, and chitin oligosaccharide from Yaizu Suisankagaku. epiBL was purchased from Xiamen Topusing Chemical. LiCl, bikinin, brassinazole and GA were purchased from Sigma (St Louis, MO, USA). Paclobutrazol was purchased from Duchefa (Haarlem, NL). Uniconazole was purchased from Sigma.

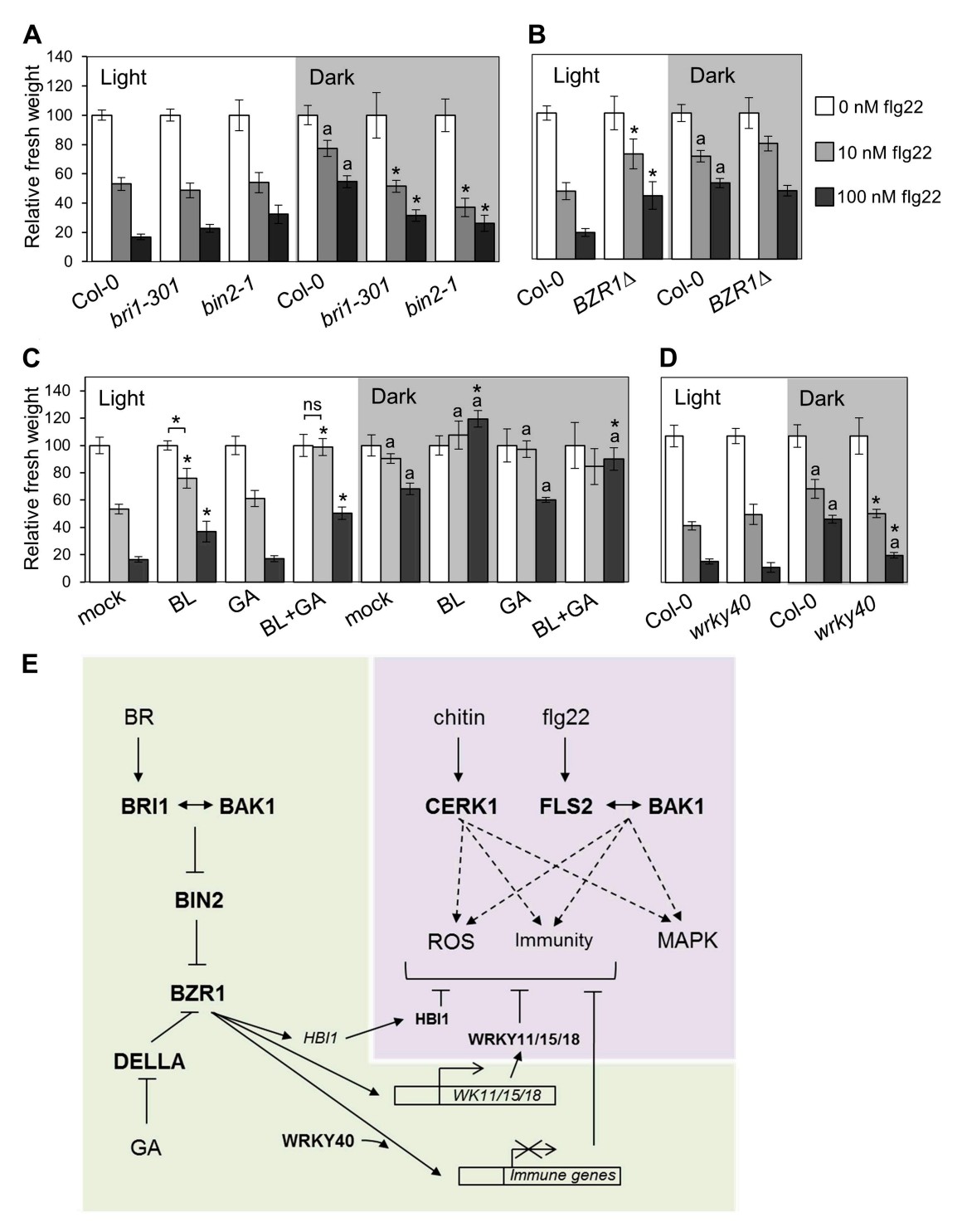

**Figure 4**. Activation of BR signaling and BZR1 prioritizes growth over immunity in the dark. (**A**) and (**B**) Relative seedling growth inhibition of 10-day-old (**A**) Col-0, *bri1-301* and *bin2-1* or (**B**) Col-0 and *BZR1Δ* seedlings induced by increasing concentrations of flg22 in either light or dark. (**C**) Relative seedling growth inhibition of 10-day-old Col-0 seedlings grown on medium supplemented or not with BL (1 µM), GA (1 µM), BL+GA (1 µM + 1 µM) or mock solution in light or dark. (**D**) Relative seedling growth inhibition of Col-0 or *wrky40* seedlings induced by increasing concentrations of flg22 in either light or dark. Bars represent SE of n = 16 (**A**, **B** and **D**) or n = 8 (**C**) Asterisks indicate a statistically significant difference compared to Col-0 in the same condition (light or dark and same concentration of flg22), according to a Student's *t*-test (p<0.05); 'a' indicates a statistically significant difference compared to the same genotype/treatment and concentration of flg22 in light, according to a Student's *t*-test (p<0.05). All experiments were repeated at least three times with similar results.
*Figure 4. Continued on next page*

*Figure 4. Continued*

Values are relative to Col-0 (**A**, **B** and **D**) or mock-treated seedlings (**C**) (set to 100). Absolute values of these experiments are shown in *Figure 4—figure supplement 3*. (**E**) Schematic model depicting the BZR1-mediated inhibition of PTI. Upon BR- and DELLA-dependent activation, BZR1 induces the expression of negative regulators of PTI, such as *WRKY11*, *WRKY15*, *WRKY18*, or *HBI1*. In addition, BZR1 also inhibits the expression of immune genes, acting cooperatively with WRKY40 and possibly other WRKYs. Ultimately, the BZR1-mediated changes in transcription would lead to the suppression of PTI signaling. The PTI signaling pathway is shadowed in violet; the BR signaling pathway is shadowed in green.

The following figure supplements are available for figure 4:

**Figure supplement 1**. The BR-mediated suppression of seedling growth inhibition in the dark requires GA synthesis.

**Figure supplement 2**. Phenotype of the light- or dark-grown seedlings used in the seedling growth inhibition assays (*Figure 4* and *Figure 4—figure supplement 1*).

**Figure supplement 3**. Absolute fresh weight values of seedling growth inhibition assays.

## ROS assays

The measurement of ROS generation was performed as described in *Albrecht et al. (2012)*. Leaf discs from five-week-old Arabidopsis plants were used in each experiment, as indicated in the figure legends. Total photon counts were measured over 40 min by using a high-resolution photon counting system (HRPCS218) (Photek, St Leonards on Sea, UK) coupled to an aspherical wide lens (Sigma Imaging, Welwyn Garden City, UK).

## Protein extraction and IP experiments

Protein extraction and immunoprecipitation of Arabidopsis was performed as described in *Schwessinger et al. (2011)*. Arabidopsis mesophyll protoplasts were prepared from 4 to 5-week-old plants, transfected with the indicated constructs and incubated for 16 hr prior to BL treatment. Protein extraction of *N. benthamiana* was performed as described in *Schwessinger et al. (2011)*; immunoprecipitations were performed using the μMACS GFP Isolation Kit (Miltenyi Biotec, Church Lane Bisley, UK), following the manufacturer's instructions. In *N. benthamiana*, BZR1-GFP was expressed from the pUb-cYFP-Dest vector (*Grefen et al., 2010*); WRKY40-HA and WRKY6-HA were expressed from the pAM-PAT vector (AY436765; GeneBank). In protoplasts, WRKY40-HA was expressed from the pGWB414 vector (*Nakagawa et al., 2007*); the construct to express BZR1-GFP has been described elsewhere (*Ryu et al., 2007*). In both cases, samples were treated with 1 μM epiBL solution for 1 hr prior to protein extraction.

## MAP kinase activation assays

MAP kinase activation assays were performed as described in *Schwessinger et al. (2011)*. Phospho-p44/42 MAPK (Erk1/2; Thr202/Tyr204) rabbit monoclonal antibodies (Cell Signaling Technologies, Hitchin, UK) were used according to the manufacturer's protocol.

## RNA isolation and qPCR assays

RNA isolation was performed from ten-day-old seedling following the protocol described in *Onate-Sanchez and Vicente-Carbajosa (2008)*. First-strand cDNA synthesis was performed with the SuperScript III RNA transcriptase (Invitrogen, Paisley, UK) and oligo(dT) primer, according to the manufacturer's instructions. For qPCR reactions, the reaction mixture consisted of cDNA first-strand template, primers (5 nmol each) and SYBR Green JumpStart Taq ReadyMix (Sigma). qPCR was performed in a BioRad CFX96 real-time system. *UBQ10* was used as the internal control; expression in mock-treated Col-0 seedlings was used as the calibrator, with the expression level set to one. Relative expression was determined using the comparative Ct method (2-ΔΔCt). Each data point is the mean value of three experimental replicate determinations. Primers for *At2g17740* are described in *Albrecht et al. (2012)*; for *NHL10* are described in *Boudsocq et al. (2010)*; for *LOX2* are described in *Pandey et al. (2010)*; for *UBQ10* (*U-box*) are described in *Albrecht et al. (2012)*.

## Seedling growth inhibition assay

Seedling growth inhibition assays were performed as described in *Nekrasov et al. (2009)*. In brief, four-day-old Arabidopsis seedlings were grown in liquid Murashige–Skoog medium containing 1%

sucrose supplemented with flg22 and the appropriate chemicals. Seedlings were weighed between 6 and 10 days after treatment.

## Bacterial infections

Induced resistance assays were performed as described previously (*Zipfel et al., 2004*). In brief, water or a 1 µM flg22 solution were infiltrated with a needleless syringe into leaves of four-week-old Arabidopsis plants 24 hr prior to bacterial inoculation (*Pto* DC3000, $10^5$ cfu/ml). Bacterial growth was determined 2 days after inoculation by plating serial dilutions of leaf extracts.

Spray inoculation of *P. syringae* pv. *cilantro* (*Pci*) 0788-9 was performed as described in *Schwessinger et al. (2011)*. In brief, bacteria were grown in an overnight culture in LB medium, cells were harvested by centrifugation, and pellets were re-suspended to OD600 = 0.02 in 10 mM $MgCl_2$ with 0.04% Silwet L-77. Bacterial suspensions were sprayed onto leaf surfaces and plants were kept uncovered. Bacterial growth was determined 3 days after inoculation by plating serial dilutions of leaf extracts.

## Meta-analysis

Functional enrichment analyses of the Biological Process ontology were performed using VirtualPlant (*Katari et al., 2010*). Functional enrichment analysis of the Molecular Function ontology was performed using the Classification SuperViewer tool of the Bio-Array Resource for Arabidopsis Functional Genomics, BAR (*Toufighi et al., 2005*). Promoter analyses were performed using Athena (*O'Connor et al., 2005*).

## Acknowledgements

RL-D is supported by a postdoctoral fellowship from Fundación Ramón Areces; APM is supported by a postdoctoral fellowship from the Federation of European Biochemical Societies. We thank Lena Stransfeld and the horticultural service at the John Innes Centre for excellent technical assistance, Yasuhiro Kadota for technical advice, and Christine Faulkner and all members of the Zipfel laboratory for fruitful discussions and helpful comments. We thank Zhiyong Wang, Joanne Chory, Ildoo Hwang, Dominique Roby, Eugenia Russinova, Xinnian Dong, Kang Chong, Zhiwiang Chen, Jun-Xian He, Jonathan Jones and David Guttman for sharing biological materials, and Sacco de Vries and Ben Scheres for excellent comments on the manuscript.

## Additional information

### Funding

| Funder | Grant reference number | Author |
| --- | --- | --- |
| The Gatsby Charitable Foundation | | Cyril Zipfel |
| UK Biotechnology and Biological Sciences Research Council | BB/G024936/1; BB/G024944/1 | Cyril Zipfel |
| Deutsche Forschungsgemeinschaft | SFB 670 | Imre E Somssich |

The funders had no role in study design, data collection and interpretation, or the decision to submit the work for publication.

### Author contributions

RL-D, Conception and design, Acquisition of data, Analysis and interpretation of data, Drafting or revising the article; APM, FB, CS, Conception and design, Acquisition of data, Analysis and interpretation of data; IES, Conception and design, Drafting or revising the article, Contributed unpublished essential data or reagents; CZ, Conception and design, Analysis and interpretation of data, Drafting or revising the article

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
