## [Decision Letter]

Thank you for sending your work entitled “The transcriptional regulator BZR1 mediates trade-off between plant innate immunity and growth” for consideration at *eLife*. Your article has been favorably evaluated by a Senior editor, Detlef Weigel, and 3 reviewers, one of whom, Thorsten Nürnberger, is a member of our Board of Reviewing Editors.

The Reviewing editor and the other reviewers discussed their comments before we reached this decision, and the Reviewing editor has assembled the following comments to help you prepare a revised submission. 

The manuscript reports on a molecular mechanism underlying opposing plant physiological programs, such as growth and immunity. This is an important yet incompletely understood problem in plant biology. The authors have convincingly demonstrated that two factors, BZR1 and WRKY40, play a crucial role in brassinolide-mediated suppression of immunity. The authors should, however, respond to the referees’ issues listed below prior to final acceptance of their work: 

1) BL treatment reverts PAMP-triggered SGI only partially, but never fully (see Figure 4). Likewise, constitutively active BZR also results in partial SGI complementation only, suggesting that other factors than BZR (and brassinolide) are implicated in regulation of this trade-off. Thus, calling BZR a master regulator is not fully justified. Similarly, there is the caveat that seedling growth might not always be regulated in the same way as overall vegetative growth. The authors may therefore provide more adequate statements throughout the text. 

2) The synergistic effect of GA and BL (shown in Figure 4 and its figure supplements) is not fully convincing. In fact, in a few cases restoration of SGI by combined hormone treatment seems to be even smaller than that observed with BL alone. As PAC inhibitor specificity is not really clear, it is suggested that the authors either analyze appropriate BL/GA-deficient mutants for a synergistic effect or, alternatively, opt to omit the GA part, which would weaken an otherwise very strong paper.

3) The long time of pre-incubation with BL (8 hrs) before flg22 treatment raises some concern as it is expected that the system has already shifted to a new steady state equilibrium. Here, the authors have to show that the brassinosteroid signaling is still active at this point and has not been reset through feedbacks. For instance, the authors could check the BZR phosphorylation status in a time course until flg22 challenge, or nuclear target gene transcription.

4) The authors state that PTI suppression might be particularly important during rapid growth phases, such as de-etiolation. Somewhat surprising here is that they then use fresh weight to measure growth. Given that brassinosteroids mainly regulate cell elongation and considering that during seedling establishment, cell expansion is the by far dominant growth mechanism, it is surprising that the authors have not considered determining hypocotyl elongation. This is an even more rapid growth phase, as compared to the authors’ fresh weight measurements at 10 days after germination. As such experiments can be rapidly conducted it should be tested experimentally whether such data match with those obtained by fresh weight measurements. 

5) The manuscript would substantially benefit if the authors could provide more direct evidence that BZR1 and WRKY40 indeed cooperate in gene regulation. For instance, the authors could conduct relatively straightforward protoplast transfection experiments with BZR1 and WRKY40 alone or together as effectors, and a *LOX2* promoter construct as a reporter. Such an experiment would strongly support the finding of a biologically relevant BZR1-WRKY40 interaction. Alternatively, chromatin-IP assays would be suited to prove that both WRKY40 and BZR1 are associated with the *LOX2* promoter. 

6) The finding that WRKY40 participates in the PTI inhibition is important. While BZR1 activation leads to reduced PTI responses and disease resistance to bacteria, it is not clear if WRKY40 also negatively impact disease resistance. An flg22-protection assay should be performed here.

---

## [Author Response]

*1) BL treatment reverts PAMP-triggered SGI only partially, but never fully (see*
Figure 4*). Likewise, constitutively active BZR also results in partial SGI complementation only, suggesting that other factors than BZR (and brassinolide) are implicated in regulation of this trade-off. Thus, calling BZR a master regulator is not fully justified. Similarly, there is the caveat that seedling growth might not always be regulated in the same way as overall vegetative growth. The authors may therefore provide more adequate statements throughout the text*.

We have now amended the text to take into account these remarks.

*2) The synergistic effect of GA and BL (shown in*
Figure 4
*and its figure supplements) is not fully convincing. In fact, in a few cases restoration of SGI by combined hormone treatment seems to be even smaller than that observed with BL alone. As PAC inhibitor specificity is not really clear, it is suggested that the authors either analyze appropriate BL/GA-deficient mutants for a synergistic effect or, alternatively, opt to omit the GA part, which would weaken an otherwise very strong paper*.

Figure 4 shows that combined treatment of BL+GA in the light has a more potent effect than BL or GA alone. In the dark, SGI is already partially suppressed, and this effect remains after exogenous application of the hormones. Thus, we do not fully understand the comment about the additive effect between the hormones being unclear. We have now added statistical analysis of Figure 4 to make this point clearer.

Regarding the specificity of PAC, we have now added results (new Figure 4—figure supplement 1) obtained after treatment with another GA biosynthesis inhibitor, uniconazole, which nicely corroborate those previously obtained with PAC. Furthermore, we report that the BL-induced inhibition of flg22-triggered SGI is hindered in the GA biosynthetic mutant *ga1-3* (new Figure 4—figure supplement 1), further confirming that the effect imposed by BL requires GA.

*3) The long time of pre-incubation with BL (8 hrs) before flg22 treatment raises some concern as it is expected that the system has already shifted to a new steady state equilibrium. Here, the authors have to show that the brassinosteroid signaling is still active at this point and has not been reset through feedbacks. For instance, the authors could check the BZR phosphorylation status in a time course until flg22 challenge, or nuclear target gene transcription*.

Most results presented in this manuscript were obtained using transgenic or mutant lines with constitutive or increased BR responses, or during constant incubation with BL, in the case of seedling growth inhibition assays (e.g., Figure 4). In [1], we also showed that co-treatment with BL inhibited flg22-induced gene expression. In the current manuscript, we only used pre-treatment in Figure 3 (8h) and Figure 1—figure supplement 2 (90 min or 5h). Nevertheless, to address the reviewers’ concern, we have tested whether BR signaling was still active after long treatments by looking at the phosphorylation status of BZR1, and found that BR signaling is still active even after an 8-hour BL treatment (this experiment has been repeated three times with similar results) (see Figure 5 below).Author response image 1.BL-induced dephosphorylation of BZR1 is maintained after an 8-hour BL treatment. Ten-day-old transgenic Arabidopsis seedlings expressing BZR1-YFP were treated with 1μM BL or mock solution for the indicated time. Proteins were detected using an anti-GFP antibody conjugated to HRP.

*4) The authors state that PTI suppression might be particularly important during rapid growth phases, such as de-etiolation. Somewhat surprising here is that they then use fresh weight to measure growth. Given that brassinosteroids mainly regulate cell elongation and considering that during seedling establishment, cell expansion is the by far dominant growth mechanism, it is surprising that the authors have not considered determining hypocotyl elongation. This is an even more rapid growth phase, as compared to the authors’ fresh weight measurements at 10 days after germination. As such experiments can be rapidly conducted it should be tested experimentally whether such data match with those obtained by fresh weight measurements*.

We measured hypocotyl length upon flg22 treatment in light-grown seedlings and did not find any impact) (see Figure 6 below). This is different to what we observed when measuring total fresh weight, and most likely reflects the fact that hypocotyls are already extremely short in Arabidopsis seedlings grown in light. In contrast, hypocotyls elongate greatly during etiolation; yet, no impact of flg22 treatment could be observed (see Figure 6 below), which is entirely consistent with the lack of flg22 responsiveness in dark-grown seedlings reported already in this study. Therefore, hypocotyl length measurement does not appear as a suitable assay to compare the impact of BR regulation on PTI between light and dark conditions, which is one of the major aims of our study.Author response image 2.Hypocotyl length of light- or dark-grown Arabidopsis seedlings in increasing concentrations of flg22. Seedlings were grown in MS plates for four days, then transferred to liquid MS supplemented with flg22 at the indicated concentrations. Hypocotyl length was measured four days later. Bars represent SE with n=8.

*5) The manuscript would substantially benefit if the authors could provide more direct evidence that BZR1 and WRKY40 indeed cooperate in gene regulation. For instance, the authors could conduct relatively straightforward protoplast transfection experiments with BZR1 and WRKY40 alone or together as effectors, and a* LOX2 *promoter construct as a reporter. Such an experiment would strongly support the finding of a biologically relevant BZR1-WRKY40 interaction. Alternatively, chromatin-IP assays would be suited to prove that both WRKY40 and BZR1 are associated with the* LOX2 *promoter*.

We now provide additional results to support the reported BZR1-WRKY40 protein interaction (Figure 3). However, while we could previously show that *LOX2* transcript levels were reduced by BL treatment in a WRKY40-dependent manner in seedlings (Figure 3 in the previous version) (a result that is reproducible), we then encountered difficulties studing *LOX2* gene regulation in other systems (e.g., protoplasts or *N. benthamiana*), systems that would be required to perform the suggested experiments in the allocated time. Furthermore, we requested previously published materials, such as *LOX2::GUS* and *LOX2::LUC* constructs or transgenic lines; but, unfortunately, these tools either are not available anymore or can no longer be used due to severe silencing of both reporter constructs. While the Kombrink lab is currently generating these tools again, we also considered making a *LOX2::LUC* construct ourselves. However, the expression of this reporter transgene has been previously reported to be quasi undetectable in the absence of exogenous MeJA treatment (Jensen et al., Plant J 2002), which would make the suggested experiment (i.e., testing whether BZR1+WRKY40 inhibit LOX2 expression more potently than BZR1 or WRKY40 alone) extremely technically challenging. For these different reasons, we have decided to omit the results on LOX2 expression from the revised manuscript. Nevertheless, the results provided still convincingly show that BZR1 acts together with WRKY40 both genetically (Figure 3) and biochemically (Figure 3) to regulate the BR-mediated inhibition of PTI. The exact mechanisms underlying the latter regulation will be within the scope of future studies.

*6) The finding that WRKY40 participates in the PTI inhibition is important. While BZR1 activation leads to reduced PTI responses and disease resistance to bacteria, it is not clear if WRKY40 also negatively impact disease resistance. An flg22-protection assay should be performed here*.

Our previous data showed that *wrky40* plants were more resistant to *Pto* DC3000 (Figure 4—figure supplement 1), indicating that WRKY40 is a negative regulator of immunity. This is consistent with other reports (Xu et al., Plant Cell 2006; Pandey et al., Plant J 2010; Schon et al., MPMI 2013; Brotman et al., PLOS Pathog 2013; Schweizer et al., Front Plant Sci 2013). When we measured flg22-induced resistance, we observed that the flg22 treatment could not further decrease *Pto* DC3000 titers in *wrky40* plants (Figure 7 below).Author response image 3.Flg22-induced resistance to *P. syringae* pv. *tomato* DC3000 in Col-0 and *wrky40* plants. Plants were pre-treated with 1 μM flg22 or water 24 hours prior to bacterial infiltration. Results are the average of three independent biological replicates; bars represent SE.